# ROS-Based Unmanned Mobile Robot Platform for Agriculture

**Eu-Tteum Baek [1],* and Dae-Yeong Im [2]**

1   Department of AI & Big Data, Honam University, Gwangju 62399, Korea
2   Automotive Materials & Components R&D Group, Korea Institute of Industrial Technology, Gwangju 61012, Korea; dylim@kitech.re.kr
*   Correspondence: geodo100@gmail.com

**Abstract:** While the demand for new high-tech technologies is rapidly increasing, difficulties are presented, such as aging and population decline in rural areas. In particular, autonomous mobile robots have been emerging in the agricultural field. Worldwide, huge investment is being made in the development of unmanned agricultural mobile robots; meanwhile with the development of robots, modern farms have high expectations of increased productivity. However, in the agricultural work environment, it is difficult to solve these problems with the existing mobile robot form, due to the difficulties of various environments. Typical problems are space constraints in the agricultural work environment, the high computational complexity of algorithms, and changes in the environment. To solve these problems, in this paper, we propose a method to design and operate a mobile robot platform that can be used in a greenhouse. We represent a robot type with two drive wheels along with four casters that can operate on path and rail. In addition, we propose a technology for a multi-AI deep learning system to operate a robot, an algorithm that can operate such a robot, and a VPN-based communication system for network and security. The proposed method is expected to increase productivity and reduce labor costs in the agricultural work environment.

**Keywords:** agricultural robot; wheeled mobile robot; robot operating system; VPN

## 1. Introduction

In rural areas, the proportion of elderly people keeps increasing, while the youth labor force participation rate is decreasing. As a result, rural areas, where there is insufficient workforce, have no choice but to attract manpower from outside. However, the reality is that it is difficult to recruit a new workforce. In addition, although recently, mechanization and the automation of agricultural work are progressing, efficient technology in the actual agricultural environment to replace the labor force has been lacking.

Globally, climate change is evident, and the temperature is rising. Agricultural production in particular is very vulnerable to climate change. For every 1 degree Celsius increase in global average temperature, it is predicted that, on average, global production of wheat, rice, corn, and soybean will decrease by (6.0, 3.2, 7.4, and 3.1)%, respectively [1]. Climate change has become an even greater obstacle to farmers struggling with a shortage of labor. Therefore, in rural areas, the development of advanced agricultural work robots is considered an urgent task.

To solve various problems [2–5], mobile robots nowadays are working in both the industrial and non-industrial environments. Advances in mobile robots allow for a variety of uses. Warehouse managers use autonomous mobile robots to manage and move inventory [6]. Household vacuum cleaner robots are very common today as a general consumer product [7]. Robots that can work in home environments to help older people [8] are another area currently being researched. In particular, the mobile robot will show great efficiency and effectiveness when applied to the smart farm environment. In recent years, many robots have been developed to manage, and take care of, crops [9]. They are being developed to work autonomously, with minimal or no human intervention. For the

robot to move autonomously, environmental awareness sensors, such as laser scans and depth cameras, are indispensable. Through the Simultaneous Localization and Mapping (SLAM) algorithm and route planner for map building and location recognition, the robot can autonomously drive [10].

In particular, various roles of robots are required in agricultural fields. To meet these demands, various types of robots have been developed until recently. Agricultural robot technology has been widely studied and applied in planting, aquaculture, livestock, and poultry farming. In addition, Agricultural robots are also being used in crop agriculture for phenotyping, monitoring, mapping, crop management, environmental control, etc.

Several agricultural harvesting robots have been developed. To harvest fruit, robots perform visual perception, position detection, segmentation, and 3D reconstruction, and compute spatial coordinates of objects [11–15]. Wibowo et al. have developed an end-to-end autonomous coconut harvesting robot that can automatically climb coconut trees and detect fruit through a vision system [11]. Qingchun et al. has developed a strawberry harvesting robot with picking speeds of 7.5 s per strawberry [12]. Zhao et al. used a CCD sensor to localize sweet-pepper in the image plane [13]. Barth et al. developed a selective sweet-pepper harvesting robot [14]. Hemming et al. developed another sweet-pepper harvesting robot [15]. Si et al. developed another apple harvesting robot [16].

In agricultural production, monitoring is just as important as harvesting. Various robots for monitoring are also being developed. The European Union VineRobot project has developed an autonomous robot that measures grape growth, grape yield, grape composition, and soil moisture. The platform is a 4WD Electric Monster Truck, which includes a navigation sensor, an infrared camera, and an RGB camera [17]. Rizk H et al. developed a plant health monitoring robot based on 5-Wheel and 6-chassis [18]. Rizk H et al. developed crop disease detection based on 4-wheel [19].

Robot technology is a powerful tool for realizing targeted actions in agriculture. Zhao Y et al. developed a seed sowing robot based on a 4-wheeled platform, which contains vision sensors, a lead screw, and a rotating disc [20]. Van Henten E J et al. developed a de-leafing robot based on railed vehicle [21,22]. Strisciuglio N et al. developed a trimming robot based on a lawnmower [23]. In addition, recently, research on developing a robot using a digital twin method is in progress [24].

However, there are still many difficulties in using these agricultural robots. In an indoor environment in particular, there are several problems, such as steering issues, network disconnection problems, robots that only solve specific problems, and so forth. One of the challenges of indoor environments is that the corridors in a typical greenhouse are narrow, and to grow more crops in the greenhouse, the joints and corners are designed to be very narrow. Because the vehicle with rear-wheel steering, which is most commonly used in vehicles, cannot turn in a tight and narrow space, agriculture mobile robots should not move like a four-wheeled car. Another problem is that there is the possibility of communication cut-off due to the steel frame, vinyl, and glass in the smart farm. If communication is suddenly cut off, there is a high risk that the robot will malfunction. Yet another problem is that it is difficult to operate in real-time due to the high computational complexity of deep learning or robotic algorithms.

To solve the problems that occur in the unmanned agricultural work robot of the smart farm, a multi-AI onboard system is built to calculate the deep learning algorithm in real-time to enable the stable use of the autonomous system. In addition, to overcome the steering issue, we designed a new autonomous mobile robot with 2WD chassis. Section 2 explains the autonomous mobile robot platform. Section 3 introduces a mobile robot system based on ROS. Section 4 shows the experimental results. Finally, Section 5 concludes the paper.

## 2. Autonomous Mobile Robot Platform

### 2.1. Analysis of the Greenhouse Environment

The proposed robot is intended to be applied to cutting-edge smart farms. For the conceptual design of the control robot, analysis is conducted on a paprika farm. The research target is the paprika smart farm in the form of an interlocking glass greenhouse in South Korea. The horticultural environment is standardized as shown in Figure 1, and the internal structure of the facility consists of a concrete passage, a hot water pipe for heating, and a bed for growing crops. Table 1 represents the specification of the smart farm. Although the corridor is not narrow to allow farmers to work, it is too narrow for the robot to move. Even the smart farms that we investigated are large in scale in South Korea. Therefore, it is necessary to design a robot that can load and transport a sufficient quantity of crops, and that can move freely, even in a tight space.

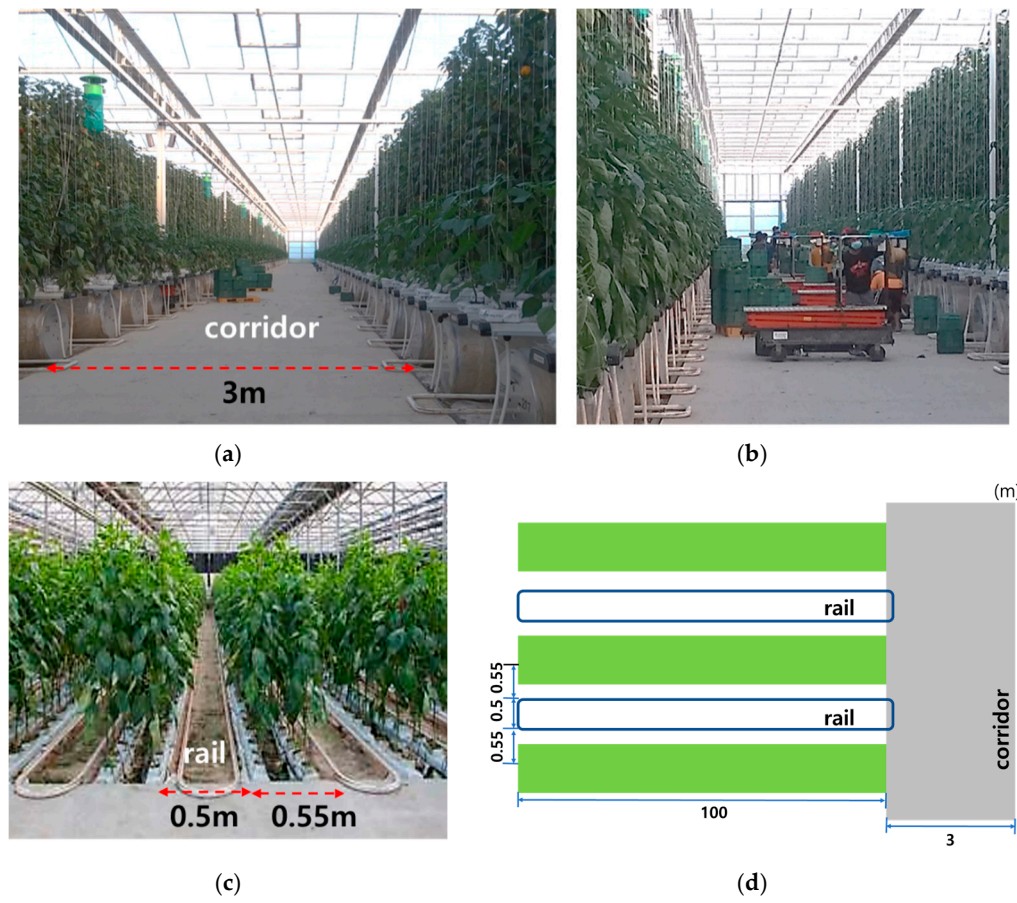

**Figure 1.** The greenhouse environment. (**a**) corridor; (**b**) corridor with machines; (**c**) rails; (**d**) top view of smartfarm.

**Table 1.** Specification of the greenhouse.

| Parameter | Value |
|---|---|
| Cultivation area | 24,300 m$^2$ |
| Length of rail | 100 m |
| Number of rails | 150 ea. |
| Width of the corridor | 3 m |

### 2.2. WD Mobile Robot Chassis

We established a design strategy based on the optimization of robot mobility, transportation, and agricultural work. First, design specifications, such as load and working

area, are reviewed according to purposes and requirements. Second, we select the hardware specifications according to the conceptual design, design the basic structure of the robot, and select the main hardware parts accordingly. Third, detailed design evaluates the capacity and characteristics of major parts by 3D modeling and manufacturing the detailed structure of the control robot. The proposed robot can work while moving along each ridge and corridor in facility cultivation areas, such as glass greenhouses and smart farms. In addition, since it runs using the hot water pipe for heating installed in each ridge, the maximum load of the robot should be designed and manufactured within 300 kg, considering the allowable load of the hot water pipe for heating.

To move freely in narrow space, we design a prototype of a 2WD mobile robot to conduct agriculture work. The advantage of the 2WD mobile is that it can turn around near narrow passages. The mobile robot chassis serves as the main body in the construction of the robot, as represented in Figure 2:

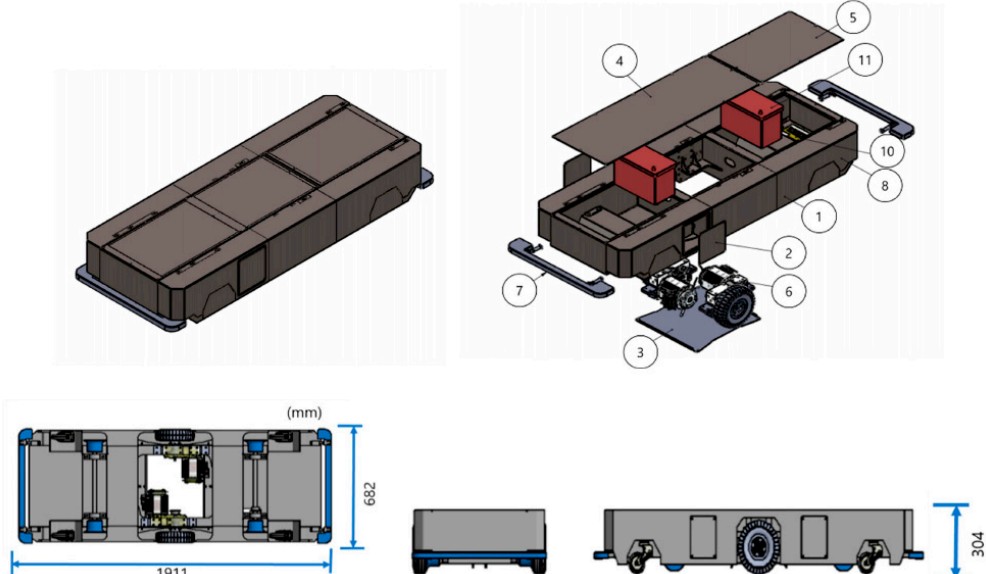

**Figure 2.** Chassis of the 2WD mobile robot model. (1) frame, (2) cover, (3) cover, (4) cover panel, (5) cover panel, (6) motor, (7) bumper, (8) slide block, (10) sensor cover, (11) battery.

The chassis is designed in such a way as to support all components of the mobile robot, and to withstand the load of 350 kg. In addition, it is designed to be operated together with the smart farm rail and passage, as shown in Figure 3. The mobile robot normally passes through the corridor, and moves on pipe rails in the facility when performing tasks, such as transplanting, planting, and leaf-cutting. To perform the tasks, the robot chassis may also be added to other components, such as a vacuum system, lift table system, or manipulator. For driving on rails, we use a method of driving the rail wheels by connecting a chain to the motor, as represented in Figure 4.

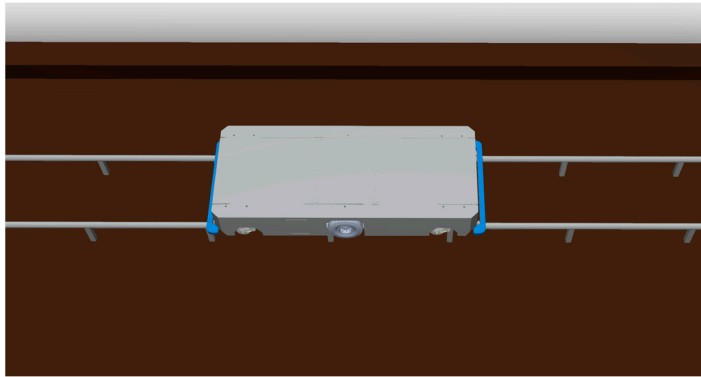

**Figure 3.** A mobile robot that moves along smart farm rails and passageways.

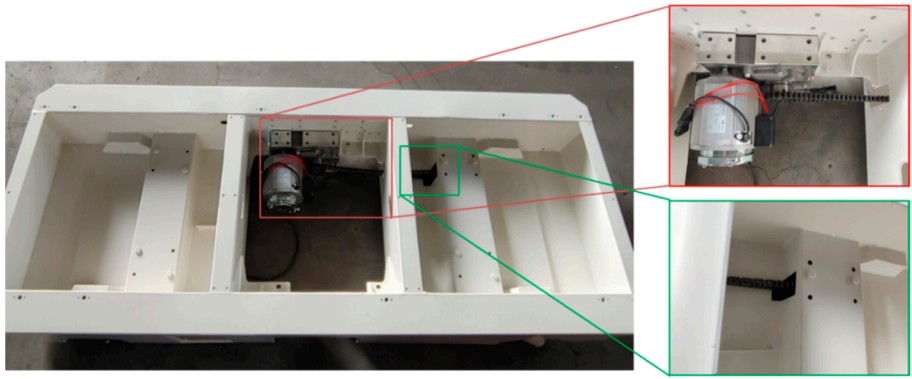

**Figure 4.** Rail wheel drive using chain belt type.

*2.3. Sensors and System Setup*

We configure a multi-on-board system for the operation of the robot in real-time. As shown in Figure 5, jetson1 is responsible for sensing data or calculating or analyzing data, or controlling the robot. Jetson2 controls additionally used robotic arms and cleaning machines. For autonomous movement, it is necessary to install sensors for the process of detecting events and quantitative mapping for further processing of changes in the surrounding environment. Therefore, we install Lidar, a camera, and an encoder to obtain data. The camera mainly detects objects and the surrounding environment. Lidar is used to map the smart farm terrain. The encoder is used to detect the position of the robot. An LTE router is also installed for communication between the server and robot.

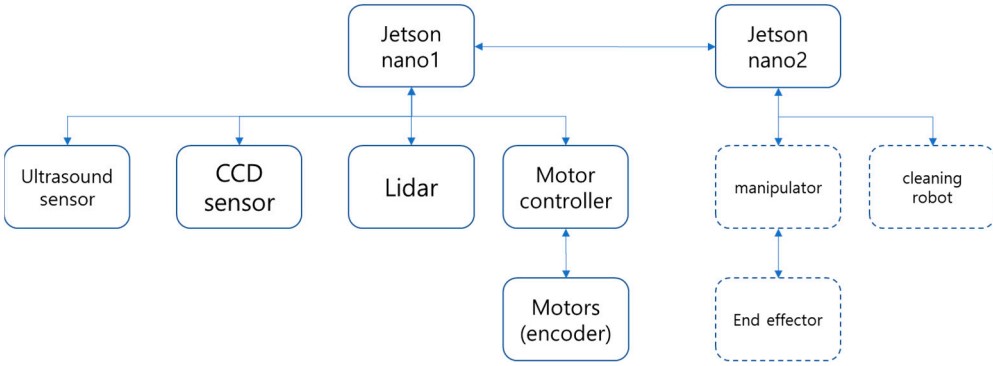

**Figure 5.** System structure.

## 3. Mobile Robot System

### 3.1. ROS Based Mobile Robot System

ROS (Robot Operating System) is a middleware system for controlling robot components from a PC. ROS is usually installed on the Operating System (OS). The ROS system consists of several independent nodes, each of which operates by communicating with other nodes using a publish/subscribe messaging model. Our robot control system employs NVidia Jetson nano embedded board as the computation module with Ubuntu 18.04 operation system and ROS Melodic. To facilitate communication between processes, develop a robot with a new ROS version, and allow easy integration of a wide range of tools and algorithms, we employ Robot Operating System (ROS) [25,26]. ROS is a middleware that provides a message passing framework and a large number of tools and libraries for robot development. The middleware, ROS, is not a complete operating system, but rather an abstraction layer or meta-operating system that runs on top of Ubuntu [27]. In addition, we implement ROS through VPN. A virtual private network (VPN) is an encrypted connection from the device to network over the Internet. Encrypted connections help to securely transmit confidential data. They prevent unauthorized persons from hacking into the user's traffic, and allow users to quickly perform tasks remotely [28]. Figure 6 represents the proposed mobile robot system using ROS through VPN. The jetson nano board is connected with motor controller FIM2360 via USB to Serial. FIM2360 motor driver, which can control up to two DC motors. FIM2360 sends current values and status such as temperature and battery to the jetson nano board. In addition, by the distribution and processing multi-jetson nano board, autonomous driving and task control can be processed. In this study, two jetson nano boards are used; but in the future, if other devices such as cleaning robots or manipulators are combined, additional boards may be added.

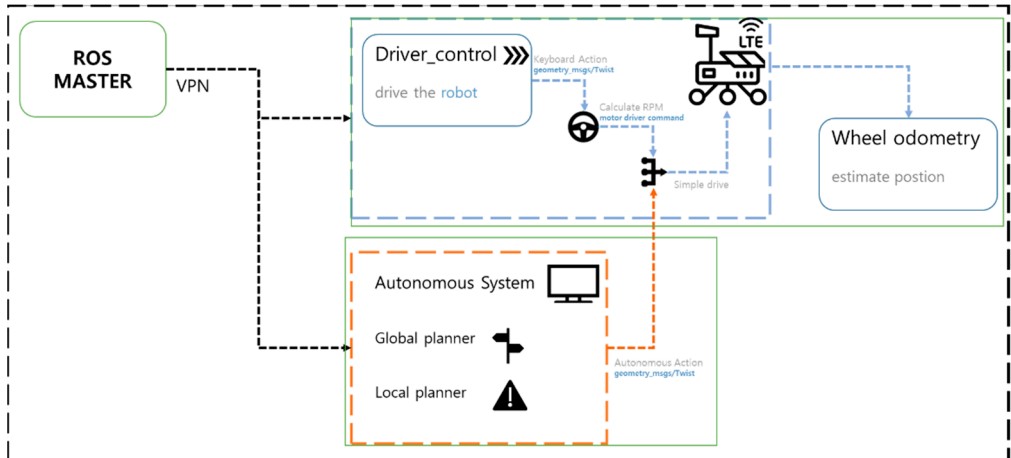

**Figure 6.** Proposed mobile robot system.

Table 2 presents detailed information about our mobile robot platform. The proposed system contains Ubuntu 18.04 operation system and ROS Melodic. The software is written in python3.

**Table 2.** Hardware and software specification of the mobile robot platform.

| Parameters | Configuration |
| --- | --- |
| Chassis | 2WD mobile robot |
| Motor controller | FIM2360 |
| Battery | 12 V $\times$2 |
| Motor | AC Induction Motor (24 V) $\times$2 |
| Board | Jetson nano board $\times$2 |
| Router | CNR-L500 (LTE) |
| OS | Ubuntu 18.04 |
| Camera | YOITCH webcam |
| Lidar | Velodyne vlp-16 |
| ROS | Melodic |
| Program language | Python3 |

### 3.2. Mobile Robot Control

Our mobile robot uses a differential drive mechanism. The robot consists of two drive wheels mounted on a common axle and each wheel can be independently driven forward or in reverse. To control the DC motors in the robot operating system (ROS), we create a custom motor drive node that subscribes to the cmd_vel topic, and publishes the base_odom topic. Our motor drive node receives cmd_vel topic from other processes. cmd_vel topic is a type of geometry_msgs/Twist. A twist is composed of a 3D linear vector (x, y, z), and a 3D angular vector (x, y, z). Values of RPM parameters are transmitted to the motor controller FIM2360 to operate motors. Thus, we are required to convert Twist to RPM with the following equation [29]:

$$RPM_l = s \left( x_{linear} - z_{angular} \times \frac{W_{robot}}{2} \right)$$
$$RPM_r = s \left( x_{linear} + z_{angular} \times \frac{W_{robot}}{2} \right)$$

(1)

where $RPM_l$ and $RPM_r$ are the RPM values of the left motor and right motor, respectively, $s$ indicates a scale factor, $x_{linear}$ is the x value of linear vector, $z_{angular}$ is the z value of angular vector, and $W_{robot}$ is the length between the left wheel and right wheel. Equation (1) is a formula that allows the robot to rotate its rolling motion around the ICC (Instantaneous Center of Curvature).

### 3.3. Cam–Lidar Calibration

To use heterogeneous cameras in one system, we extract extrinsic parameters between each camera and Lidar sensors [30]. This is to use the information of the object predicted by the camera in the lidar sensor. Camera parameters include intrinsic, extrinsic, and distortion coefficients. Before estimating the extrinsic parameters between the camera and Lidar sensor, we need to extract the intrinsic parameter of the camera [31]. The intrinsic parameters are defined as:

$$\begin{bmatrix} f_x & 0 & 0 \\ s & f_y & 0 \\ c_x & c_y & 1 \end{bmatrix}$$

(2)

where $c_x$ and $c_y$ are the optical centers in pixels, and $f_x$ and $f_y$ are the focal lengths in pixels. $s$ is a skew coefficient, which if the image axes are not perpendicular, is non-zero. It calculates camera parameters based on feature points extracted from several images of the checkerboard. Figure 7 represents checkerboard images captured from different viewing angles to obtain the camera parameters. We calculate the 3D positions of corresponding feature points from captured checkerboard images. Then we find camera parameters that minimize the projection errors when the 3D points project to the checkerboard images.

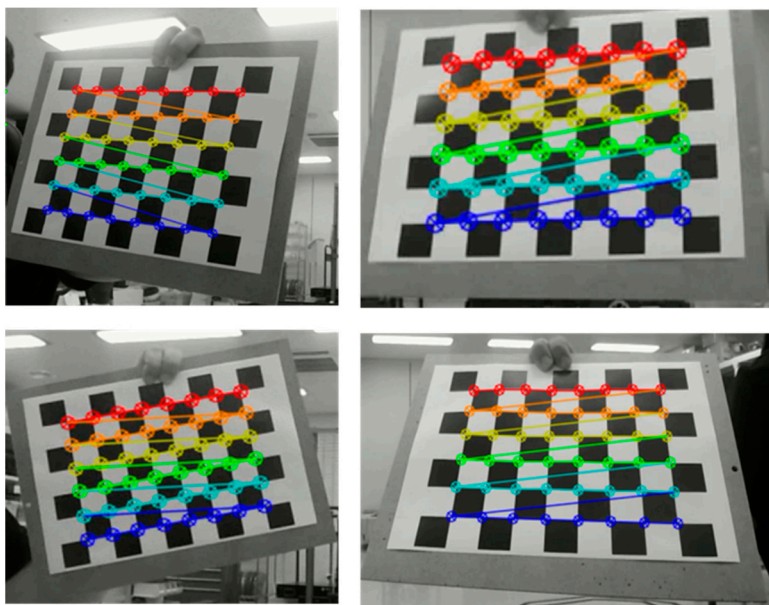

**Figure 7.** Checkerboard images for camera calibration.

To estimate the extrinsic matrix, we solve the perspective-*n*-point system. Perspective-n-point [32] is the problem of estimating the pose of a calibrated camera given a set of *n* 3D points of the world and the corresponding 2D projection of the image. When there are outliers in the point correspondence set, PnP is error-prone. Therefore, RANSAC can be used to make the final solution for the camera pose more robust against outliers [33]. We employ PnP with a RANSAC algorithm to estimate rotation and translation transforms between the camera and the LiDAR. We pick the corresponding points by selecting the four corner points of the checkerboard in both the camera and the LiDAR frames, as shown in Figure 8. The calibrated extrinsic parameter is as follows:

$$[\mathrm{R}|\mathrm{t}] = \begin{bmatrix} r_{1,1} & r_{1,2} & r_{1,3} & t_1 \\ r_{2,1} & r_{2,2} & r_{2,3} & t_2 \\ r_{3,1} & r_{3,2} & r_{3,3} & t_3 \end{bmatrix} \tag{3}$$

where *R* is the rotation matrix, and *t* is the translation matrix.

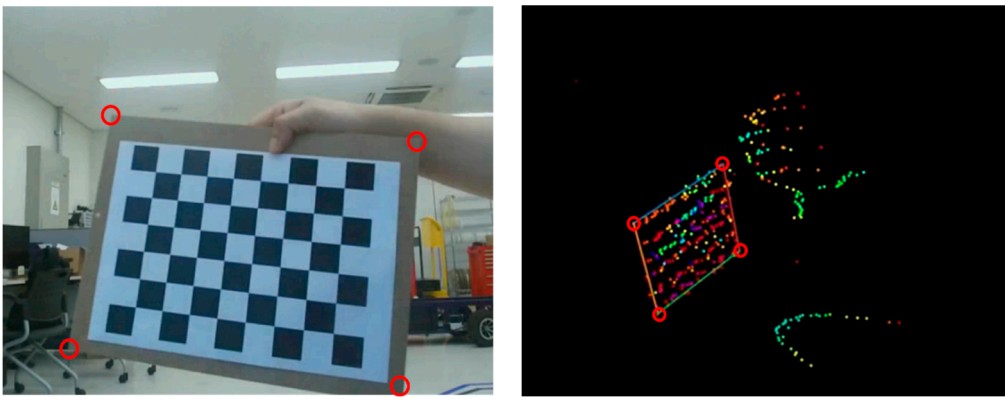

**Figure 8.** Corresponding points by selecting the four corner points of the checkerboard.

### 3.4. Virtual Private Network

OpenVPN is software that can be used to create a Virtual Private Network (VPN) over the Internet [34]. In this setup, an OpenVPN access server is set up on the lab computer, allowing the robot to connect to the lab network through a gateway as shown in Figure 9.

This set-up is a so-called Site-to-Site configuration, and allows the robot network and lab network to act virtually as a single network. The basic principles of data transfer in VPN site-to-site networks in this study are as follows. The computer (Jetson board) on the robot network asks to connect to a server on the lab network. This request is routed through the robot's LTE network to the gateway. The gateway then sends a request to the lab's access server through an encrypted VPN tunnel. The access server decrypts the message, and sends it to the target computer (or server) over the virtual network. When the target computer responds, the response goes through the virtual network's routing table to the access server. Conversely, the access server acts as a gateway to the lab's local area network, and sends responses through an encrypted VPN tunnel to the robot's gateway computer. Finally, the gateway forwards the response to the originating computer.

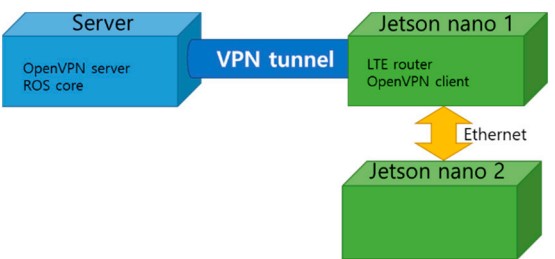

**Figure 9.** Virtural private network.

## 4. Results

This section represents the specification of the mobile robot and experimental results. Figure 10 represents the actual hardware prototype built to implement the system in Figure 5. We wire two 12 V batteries in series to make 24 V. Switch-mode power supply converts (24 to 5, 9, and 12) V to power various boards and sensors. We connect the Jetson board to the FIM2360, router, and sensors to control the robot. For agricultural work, the robot must withstand a sufficient load. Therefore, we tested how much the robot can withstand when the load increases for each part, as shown in Figure 11. In the case of a centralized load, Table 3 shows that the caster and robot are not abnormal, even at 450 kg. However, in the case of a front concentrated load, Table 4 shows that at more than 400 kg, the caster collapsed. To use our robot platform, it is safe to use less than 370 kg when the load is concentrated on one side.

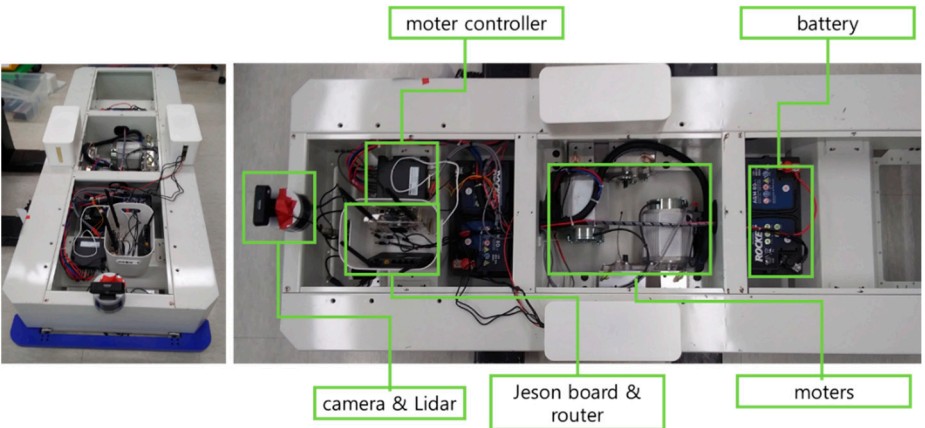

**Figure 10.** Agriculture mobile robot.

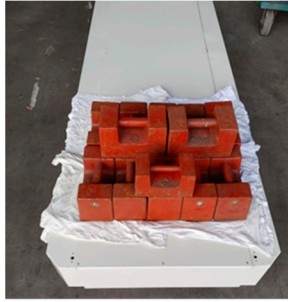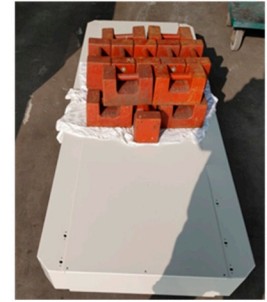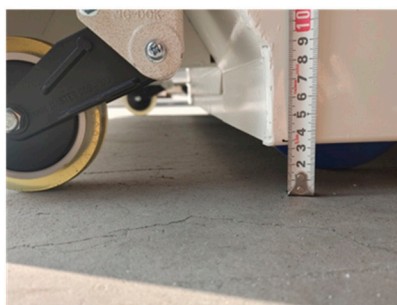

**Figure 11.** Load testing in a real environment.

**Table 3.** The case of a centralized load.

| Weight [kg] | Gap between the Ground and the Robot [mm] |
| --- | --- |
| 0 | 35 |
| 50 | 35 |
| 100 | 35 |
| 150 | 35 |
| 200 | 35 |
| 250 | 35 |
| 300 | 35 |
| 350 | 35 |
| 400 | 35 |
| 450 | 35 |

**Table 4.** The case of a front concentrated load.

| Weight [kg] | Gap between the Ground and the Robot [mm] |
| --- | --- |
| 0 | 35 |
| 50 | 35 |
| 100 | 35 |
| 150 | 35 |
| 200 | 35 |
| 250 | 35 |
| 300 | 35 |
| 350 | 35 |
| 400 | 34.8 |
| 450 | x |

To experiment with node behavior, we utilize a mobile-based controller to publish a speed command (cmd_vel) to the mobile robot. We have confirmed that the robot operates accurately on the ground and rails. To qualitatively evaluate the results of the calibration, the cloud point of the lidar was projected onto the image. For this, the data recorded in the Rosbag are played, and the point cloud is projected. Figure 12 shows the projection result in the ROS environment. Although a small error occurs due to the close distance, it seems that the performance of autonomous driving can be improved by recognizing objects using heterogeneous cameras.

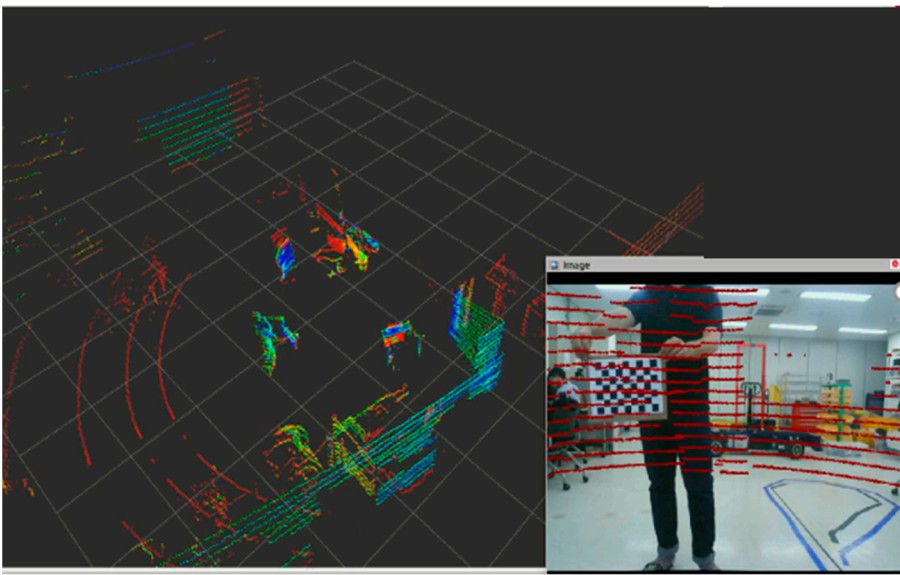

**Figure 12.** Camera-LiDAR Projection.

In general, agricultural machines are often more than 2 m in length. Therefore, 4wd-based agricultural machines cannot move easily in narrow corridors. Therefore, when moving from the hallway to the next rail, a person must manually rotate it using a caster. However, the proposed robot platform can rotate in place, so it can freely move corridors and rails without human assistance. Figure 13 represents our robot driving forward, backward, and rotating by the angle θ or in place.

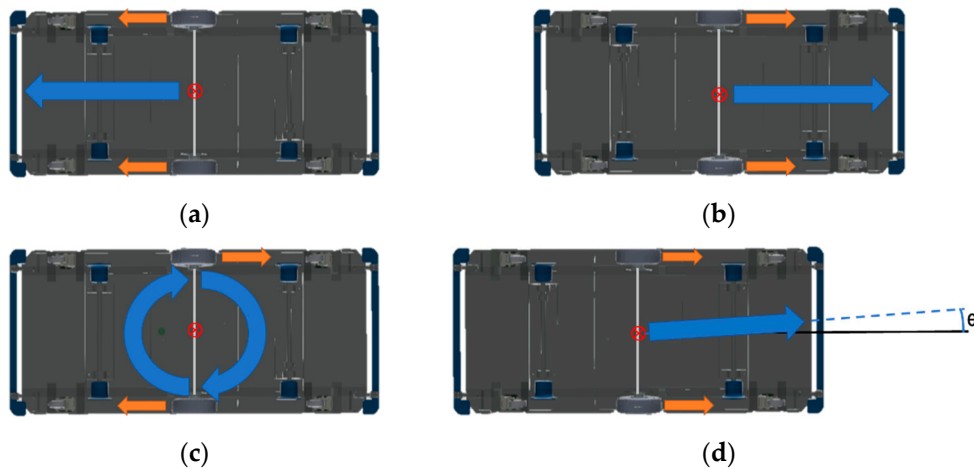

**Figure 13.** The proposed robot movement. (**a**) forward move; (**b**) backward move; (**c**) rotation in place; (**d**) rotation by the angle θ.

## 5. Conclusions

This study proposed an ROS-based unmanned mobile robot platform for agricultural work. In smart farms, there are many problems, such as steering problems, network disconnection problems, and robots that only solve specific problems. To solve the above problems, we constructed a multi-AI on-board system based on ROS to calculate deep learning methods in real-time, and designed a new 2WD mobile robot with four casters to move freely in a narrow space. In addition, we installed a VPN to protect information and communicate quickly, and implemented calibration for heterogeneous cameras. Through them, it is possible to freely move in a narrow space and perform agricultural work, and data fusion between heterogeneous cameras is possible. Consequently, it is expected that

the proposed method will be used for various agricultural work by overcoming difficulties in the agricultural environment. In future work, the robot may also be added to other components, such as a vacuum system to clean the greenhouse, a lift table system to raise or lower goods and/or persons, or a manipulator to make a transplant and to cut out the leaf.

**Author Contributions:** Conceptualization, E.-T.B. and D.-Y.I.; Data curation, E.-T.B.; Formal analysis, E.-T.B.; Funding acquisition, D.-Y.I.; Investigation, E.-T.B.; Methodology, E.-T.B.; Project administration, D.-Y.I.; Resources, E.-T.B.; Software, E.-T.B.; Validation, E.-T.B.; Writing—original draft, E.-T.B.; Writing—review and editing, E.-T.B. All authors have read and agreed to the published version of the manuscript.

**Funding:** This work was supported by the Korea Institute of Planning and Evaluation for Technology in Food, Agriculture and Forestry (IPET) and the Korea Smart Farm R&D Foundation (KosFarm), through the Smart Farm Innovation Technology Development Program, funded by the Ministry of Agriculture, Food and Rural Affairs (MAFRA) and the Ministry of Science and ICT (MSIT), Rural Development Administration (RDA) (421032-04-2-SB010).

**Institutional Review Board Statement:** Not applicable.

**Informed Consent Statement:** Not applicable.

**Data Availability Statement:** Not applicable.

**Conflicts of Interest:** We declare that we have no financial and personal relationships with other people or organizations that could have inappropriately influenced our work, there is no professional or other personal interest of any nature or kind in any product, service, and/or company that could be construed as influencing the position presented in, or the review of, the manuscript entitled, "ROS-based unmanned mobile robot platform for agriculture".

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
