# Peer review of "ROS-Based Unmanned Mobile Robot Platform for Agriculture"

_applsci, doi:10.3390/app12094335_

Round 1

Reviewer 1 Report

Dear Authors:

Ros-Based Unmanned Mobile Robot Platform for Agriculture is the subject of the paper. The research has the potential to be a good paper however in the present form the paper is not easy to follow and my specific comments are given below:

Introduction Section:

Not enough papers are cited. Especially on ROS and AI based robots.

Figure 1: Instead of 4 small figures, is it possible have a single big image that shows parts of interest ? Can you add the label for the rails in the photo ?

Section 2.2.2. WD mobile robot chassis should have the section number as 2.2 only.

Figure 2 is too small to read. It can be replaced with only 1 top view and the isometric view with the larger part number labels. Can you add the overall dimensions of the robot (length, width and height)

Figure 3: Only 1 image is sufficient.

Section 3: Design selection criteria for robot chassis, controller, motors and sensors is not clearly explained.

Table 2 from the results section should be moved to the section 3.

Equation 1 demands more explanation.

ROS and deep learning system should be explained in more detail.

Results section needs significant changes. The table about load and distance from the ground is not a good measurement criteria.  The robot's performance should be explained using accuracy, cycle time, power consumption, path and position tracking, speed etc. None of this is reported.

Figure 10: Too small to read the details. The textual information in the figure should be represented as a table.

More detailed explanation is required for the role of deep learning and AI in the context of the tasks.

The paper needs considerable amount of rework and reorganization.

All the best!

Author Response

The authors would like to express many thanks to the reviewers for their invaluable comments. Based on them, we have revised the previous manuscript as follows.

Point 1: Not enough papers are cited. Especially on ROS and AI based robots.

Response 1:  We cited further relevant papers. [21-34]

Point 2:  Figure 1: Instead of 4 small figures, is it possible have a single big image that shows parts of interest ? Can you add the label for the rails in the photo ?

Response 2:  We enlarged the image, and add the labels in the images. You can find Figure 1 in line 112.

Point 3:  Section 2.2.2. WD mobile robot chassis should have the section number as 2.2 only.

Response 3:  We have corrected the part you mentioned. You can find it in line 119.

Point 4:  Figure 2 is too small to read. It can be replaced with only 1 top view and the isometric view with the larger part number labels. Can you add the overall dimensions of the robot (length, width and height)’

Response 4:  I added the width and height of the robot and made the text bigger in Fig. 2. You can fine Figure 2 in lind 136.

Point 5:  Figure 3: Only 1 image is sufficient.

Response 5:  We chose one image. You can find it in line 150.

Point 6:  Section 3: Design selection criteria for robot chassis, controller, motors and sensors is not clearly explained.

Response 6:  I added content and pictures. You can find them in line 166 (Figure 5) and in line 266 (Figure 9)

Point 7:  Table 2 from the results section should be moved to the section 3.

Response 7:  We move Table 2 from the results section to the section 3.

Point 8:  Equation 1 demands more explanation.

Response 8:  An explanation for Equation (1) has been added. You can find it in lines 204-217

Point 9:  ROS and deep learning system should be explained in more detail.

Response 9: we explained our system in more detail. You can find it in lines 204-217

Point 10:  Figure 10: Too small to read the details. The textual information in the figure should be represented as a table.

Response 10 : We have enlarged the image.

Reviewer 2 Report

This study proposed a ROS-based unmanned mobile robot platform for agricultural work. In smart farms, there are many problems, such as steering problems, network disconnection problems, and robots that only solve specific problems. To solve the above problems, we constructed a multi-AI on-board system based on ROS to calculate deep learning methods in real-time, and designed a new 2WD mobile robot with 4 casters to  move freely in a narrow space.

To solve the problems that occur in the unmanned agricultural work robot of the smart farm, a multi-AI onboard system is built to calculate the deep learning algorithm in real-time to enable the stable use of the autonomous system.

In addition, we installed a VPN to protect information and communicate quickly, and implemented calibration for heterogeneous cameras. 

The Introduction section is very short. This section should contain The actuality of the problem, the current research revision in this subject area with the allocation of unsolved parts of the general problem, and finally, formulation of the research goal.

The presented article could contain several used literature.

The article would be appropriate to explicitly indicate the scientific benefits of the present article explicitly.

In the present article, Figure 1 has poor readability of the units used and it would be appropriate to improve their graphics.

The conclusion section is quite short. This section does not reflect the entire content of the manuscript, including the analysis of the results obtained. This section should be extended.

Author Response

The authors would like to express many thanks to the reviewers for their invaluable comments. Based on them, we have revised the previous manuscript as follows.

Point 1:  The Introduction section is very short. This section should contain The actuality of the problem, the current research revision in this subject area with the allocation of unsolved parts of the general problem, and finally, formulation of the research goal.

Response 1:  We added the content of the introduction. You can find it in lines 53-79.

Point 2:  The presented article could contain several used literature.

Response 2:  We introduced similar works in the introduction. You can find them in [21-34]

Point 3:  The article would be appropriate to explicitly indicate the scientific benefits of the present article explicitly.

Response 3:  We added to content in sections 2-5.

Point 4:  In the present article, Figure 1 has poor readability of the units used and it would be appropriate to improve their graphics.

Response 4:  We enlarged the image, and add the labels in the images.

Point 5:  The conclusion section is quite short. This section does not reflect the entire content of the manuscript, including the analysis of the results obtained. This section should be extended.

 Response 5: we extended the conclusion section.

Reviewer 3 Report

Dear Authors,

I found your article very interesting, but I suggest introducing following remarks, which have to be added and fulfilled before publishing the paper:

  1. Regarding the abbreviation ROS, I would suggest to convolute it in the Introduction. This is a very popular robot program, but not everybody is familiar with it.

  1. Regarding the Introduction, I miss mentioning the digital-twin approach. I suggest to mention about it and citing proper paper regarding it e.g.: Stączek et al. “A digital twin approach for the improvement of an autonomous mobile robots (AMR’s) operating environment – A case study”, Sensors 2021, 21(23) 7830.

  1. I have a question regarding mapping. As you plan to you rails for your AGV, so why do you need mapping? The length of it can be pre-defined in ROS.

  1. As you have mentioned about the possible application of robots in collecting paprika, please explain how this collection will be organized?

  1. In Figure 2, the 11 number is missing.

  1. In Conclusions, I can’t further directions of development of your work. It is mentioned, that it is your initial paper regarding this topic, so the definition of next steps is relevant.

After improving the above described issues in the paper I’d like to give my positive opinion on signing my review report.

Author Response

The authors would like to express many thanks to the reviewers for their invaluable comments. Based on them, we have revised the previous manuscript as follows.

Point 1:  Regarding the abbreviation ROS, I would suggest to convolute it in the Introduction. This is a very popular robot program, but not everybody is familiar with it.

Response 1:  We added a description for ROS. You can find it on lines 171-175.,

Point 2:  Regarding the Introduction, I miss mentioning the digital-twin approach. I suggest to mention about it and citing proper paper regarding it e.g.: Stączek et al. “A digital twin approach for the improvement of an autonomous mobile robots (AMR’s) operating environment – A case study”, Sensors 2021, 21(23) 7830.

Response 2:  We have also added what you mentioned. [34]

Point 3:  I have a question regarding mapping. As you plan to you rails for your AGV, so why do you need mapping? The length of it can be pre-defined in ROS.

Response 3:  On the rails, we'll include robots that do transplants, planting, cleaning, and more. For this, it is important to recognize the exact location, so the process of mapping is necessary.

Point 4:  As you have mentioned about the possible application of robots in collecting paprika, please explain how this collection will be organized?

Response 4:  Our ultimate goal is to automate paprika production. To this end, the current goal is to develop an autonomous driving robot platform that can perform tasks automatically by replacing cleaning robots and robotic arms. The cleaning robot cleans the leaves and branches. The robotic arm performs transplants and pruning, etc.

Point 5:  In Figure 2, the 11 number is missing.

Response 5: We added the missing content.

Reviewer 4 Report

I have read manuscript with great attention and interest. The article deals with a topic of new robot based on ROS for agriculture. The application is interesting and I think that the article can be reconsidered after revision. In general, you have a proper academic way of referring and a good language. Unfortunately, there is very short Introduction section and Related works section.

Congratulations to the authors of the work.

Comments and suggestions:

  1. ROS distribution name "melodic" should be written as "Melodic", I think.
  2. It is necessary to expand the section on similar works. Without this, the article cannot be considered scientific. This section is very brief. It is necessary to do an in-depth analysis of similar solutions.
  3. I recommend adding a picture showing how the various electronic components work together. The idea is to make it clear where ROS is implemented and what hardware is connected to that electronic board. Something like Figure 5 but with electronics.
  4. Please make an analysis of possible improvements of your system and add it to the article.

After revision, the article can be judged again.

Author Response

The authors would like to express many thanks to the reviewers for their invaluable comments. Based on them, we have revised the previous manuscript as follows.

Point 1:  ROS distribution name "melodic" should be written as "Melodic", I think.

Response 1:  We corrected the word.

Point 2:  It is necessary to expand the section on similar works. Without this, the article cannot be considered scientific. This section is very brief. It is necessary to do an in-depth analysis of similar solutions.

Response 2:  We introduced similar works in the introduction. You can find it in lines 53-79.

Point 3:  I recommend adding a picture showing how the various electronic components work together. The idea is to make it clear where ROS is implemented and what hardware is connected to that electronic board. Something like Figure 5 but with electronics.

Response 3:  We added content and pictures as you suggested. You can find it in line 167.

Point 4:  Please make an analysis of possible improvements of your system and add it to the article.

Response 4:  We added improvements in the results section.

Round 2

Reviewer 1 Report

Dear Authors:

Thank you for uploading the revised manuscript!

The changes look good.

Reviewer 3 Report

Dear Authors,

You have introduced all remarks, which I have proposed. I can give my positive opinion about the manuscript.

Reviewer 4 Report

I think that the article can be accepted.